# Modifying Memories in a Recurrent Neural Network Unit

## Abstract

Long Short-Term Memory (LSTM) units have the ability to memorise and use long-term dependencies between inputs to generate predictions on time series data. We introduce the concept of modifying the cell state (memory) of LSTMs using rotation matrices parametrised by a new set of trainable weights. This addition shows significant increases of performance on some of the tasks from the bAbI dataset.

## 1 Introduction

In the recent years, Recurrent Neural Networks (RNNs) have been successfully used to tackle problems with data that can be represented in the shape of time series. Application domains include Natural Language Processing (NLP) (translation (Rosca & Breuel, 2016), summarisation (Nallapati et al., 2016), question answering and more), speech recogition (Hannun et al., 2014; Graves et al., 2013), text to speech systems (Arik et al., 2017), computer vision tasks (Stewart et al., 2016; Wu et al., 2017), and differentiable programming language interpreters (Riedel et al., 2016; Rocktäschel & Riedel, 2017).

An intuitive explanation for the success of RNNs in fields such as natural language understanding is that they allow words at the beginning of a sentence or paragraph to be memorised. This can be crucial to understanding the semantic content. Thus in the phrase *"The cat ate the fish"* it is important to memorise the subject (cat). However, often later words can change the meaning of a senstence in subtle ways. For example, *"The cat ate the fish, **didn't it**"* changes a simple statement into a question. In this paper, we study a mechanism to enhance a standard RNN to enable it to modify its memory, with the hope that this will allow it to capture in the memory cells sequence information using a shorter and more robust representation.

One of the most used RNN units is the Long Short-Term Memory (LSTM) (Hochreiter & Schmidhuber, 1997). The core of the LSTM is that each unit has a *cell state* that is modified in a gated fashion at every time step. At a high level, the cell state has the role of providing the neural network with *memory* to hold long-term relationships between inputs. There are many small variations of LSTM units in the literature and most of them yield similar performance (Greff et al., 2017).

The memory (cell state) is expected to encode information necessary to make the next prediction. Currently the ability of the LSTMs to rotate and swap memory positions is limited to what can be achieved using the available gates. In this work we introduce a new operation on the memory that explicitly enables rotations and swaps of pairwise memory elements. Our preliminary tests show performance improvements on some of the bAbI tasks (Weston et al., 2015) compared with LSTM based architectures.

## 2 The rotation gate

In this section we introduce the idea of adding a new set of parameters for the RNN cell that enable rotation of the cell state. The following subsection shows how this is implemented in the LSTM unit.

One of the key innovations of LSTMs was the introduction of gated modified states so that if the gate neuron $i$ is saturated then the memory $c_i(t-1)$ would be unaltered. That is, $c_i(t-1) \approx c_i(t)$

with high accuracy. The fact that the amplification factor is very close to 1 prevents the memory vanishing or exploding over many epochs.

To modify the memory, but retain an amplification factor of 1 we take the output after appling the forget and add gates (we call it $\mathbf{d_t}$), and apply a rotation matrix $\mathbf{U}$ to obtain a modified memory $\mathbf{c}_t = \mathbf{U}\mathbf{d}_t$. Note that, for a rotation matrix $\mathbf{U}^T\mathbf{U} = \mathbf{I}$ so that $\|\mathbf{d_t}\| = \|\mathbf{c_t}\|$.

We parametrise the rotation by a vector of angles

$$\mathbf{u} = 2\pi\sigma(\mathbf{W}_{\text{rot}}\mathbf{x} + \mathbf{b}_{\text{rot}}), \tag{1}$$

where $\mathbf{W}_{\text{rot}}$ is a weight matrix and $\mathbf{b}_{\text{rot}}$ is a bias vector which we learn along with the other parameters. $\mathbf{x}$ is the vector of our concatenated inputs (in LSTMs given by concatenating the input for the current timestep with the output from the previous time step).

A full rotation matrix is parametrisable by $n(n-1)/2$ parameters (angles). Using all of these would introduce a huge number of weights, which is likely to over-fit. Instead, we have limited ourselves to considering rotations between pairs of inputs $d_i(t)$ and $d_{i+1}(t)$. Exploring more powerful sets of rotations is currently being investigated.

Our rotation matrix is a block-diagonal matrix of 2D rotations

$$\mathbf{U}(\mathbf{u}) = \begin{bmatrix} \cos u_1 & -\sin u_1 & & & \\ \sin u_1 & \cos u_1 & & & \\ & & \ddots & & \\ & & & \cos u_{n/2} & -\sin u_{n/2} \\ & & & \sin u_{n/2} & \cos u_{n/2} \end{bmatrix}, \tag{2}$$

where the cell state is of size $n$. Our choice of rotations only needs $n/2$ angles.

## 2.1 RotLSTM

In this section we show how to add memory rotation to the LSTM unit. The rotation is applied after the forget and add gates and before using the current cell state to produce an output.

The RotLSTM equations are as follows:

$$\mathbf{x} = [\mathbf{h_{t-1}}, \mathbf{x_t}], \tag{3}$$
$$\mathbf{f_t} = \sigma(\mathbf{W}_f\mathbf{x} + \mathbf{b}_f), \tag{4}$$
$$\mathbf{i_t} = \sigma(\mathbf{W}_i\mathbf{x} + \mathbf{b}_i), \tag{5}$$
$$\mathbf{o_t} = \sigma(\mathbf{W}_o\mathbf{x} + \mathbf{b}_o), \tag{6}$$
$$\mathbf{u_t} = 2\pi\sigma(\mathbf{W}_{\text{rot}}\mathbf{x} + \mathbf{b}_{\text{rot}}), \tag{7}$$
$$\mathbf{d_t} = \mathbf{f_t} \circ \mathbf{c_{t-1}} + \mathbf{i_t} \circ \tanh(\mathbf{W}_c\mathbf{x} + \mathbf{b}_c), \tag{8}$$
$$\mathbf{c_t} = \mathbf{U}(\mathbf{u_t})\mathbf{d_t}, \tag{9}$$
$$\mathbf{h_t} = \mathbf{o_t} \circ \tanh(\mathbf{c_t}), \tag{10}$$

where $\mathbf{W}_{\{f,i,o,\text{rot},c\}}$ are weight matrices, $\mathbf{b}_{\{f,i,o,\text{rot},c\}}$ are biases ($\mathbf{W}$s and $\mathbf{b}$s learned during training), $\mathbf{h_{t-1}}$ is the previous cell output, $\mathbf{h_t}$ is the output the cell produces for the current timestep, similarly $\mathbf{c_{t-1}}$ and $\mathbf{c_t}$ are the cell states for the previous and current timestep, $\circ$ is element-wise multiplication and $[\cdot, \cdot]$ is concatenation. $\mathbf{U}$ as defined in Equation 2, parametrised by $\mathbf{u_t}$. Figure 1 shows a RotLSTM unit in detail.

Assuming cell state size $n$, input size $m$, the RotLSTM has $n(n+m)/2$ extra parameters, a 12.5% increase (ignoring biases). Our expectation is that we can decrease $n$ without harming performance and the rotations will enforce a better representation for the cell state.

## 3 EXPERIMENTS AND RESULTS

To empirically evaluate the performance of adding the rotation gate to LSTMs we use the toy NLP dataset bAbI with 1000 samples per task. The bAbI dataset is composed of 20 different tasks of various difficulties, starting from easy questions based on a single supporting fact (for example:

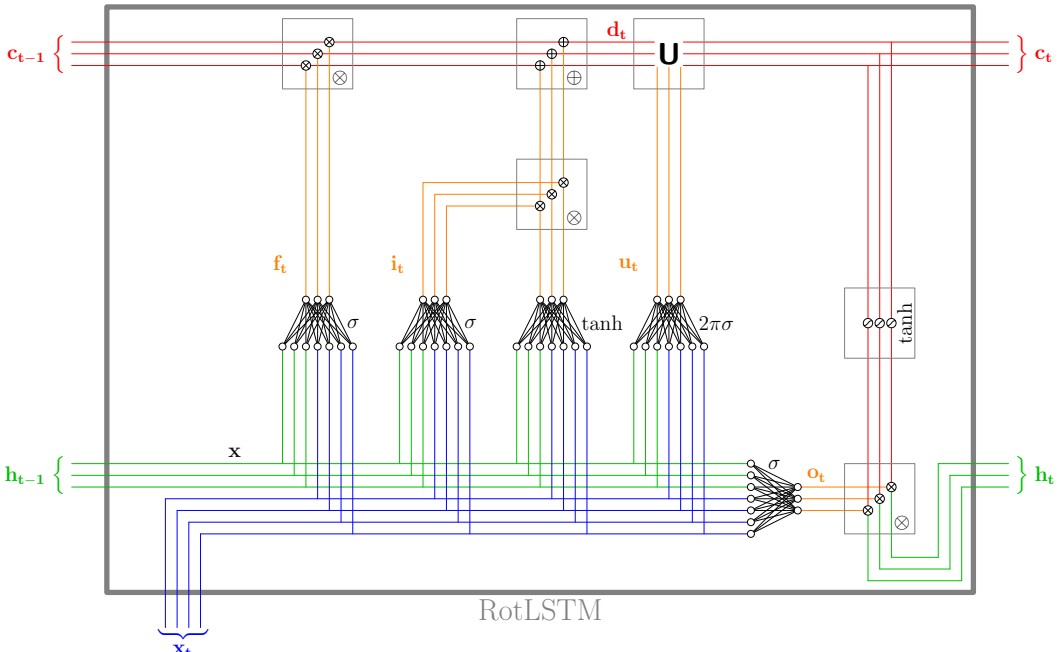

Figure 1: RotLSTM diagram. $\mathbf{x}$ is the concatenation of $\mathbf{h_{t-1}}$ and $\mathbf{x_t}$ in the diagram (green and blue lines). Note that this differs from a regular LSTM by the introduction of the network producing angles $\mathbf{u_t}$ and the rotation module marked $\mathbf{U}$. In the diagram input size is 4 and cell state size is 3.

*John is in the kitchen. Where is John? A: Kitchen*) and going to more difficult tasks of reasoning about size (example: *The football fits in the suitcase. The box is smaller than the football. Will the box fit in the suitcase? A: yes*) and path finding (example: *The bathroom is south of the office. The bathroom is north of the hallway. How do you go from the hallway to the office? A: north, north*). A summary of all tasks is available in Table 2. We are interested in evaluating the behaviour and performance of rotations on RNN units rather than beating state of the art.

We compare a model based on RotLSTM with the same model based on the traditional LSTM. All models are trained with the same hyperparameters and we do not perform any hyperparameter tuning apart from using the sensible defaults provided in the Keras library and example code (Chollet et al., 2015).

For the first experiment we train a LSTM and RotLSTM based model 10 times using a fixed cell state size of 50. In the second experiment we train the same models but vary the cell state size from 6 to 50 to assess whether the rotations help our models achieve good performance with smaller state sizes. We only choose even numbers for the cell state size to make all units go through rotations.

## 3.1 THE MODEL ARCHITECTURE

The model architecture, illustrated in Figure 2, is based on the Keras example implementation[1]. This model architecture, empirically, shows better performance than the LSTM baseline published in Weston et al. (2015). The input question and sentences are passed thorugh a word embedding layer (not shared, embeddings are different for questions and sentences). The question is fed into an RNN which produces a representation of the question. This representation is concatenated to every word vector from the story, which is then used as input to the second RNN. Intuitively, this helps the second RNN (Query) to *focus* on the important words to answer the question. The output of the second RNN is passed to a fully connected layer with a softmax activation of the size of the dictionary. The answer is the word with the highest activation.

---

[1]Available at `https://goo.gl/9wfzr5`.

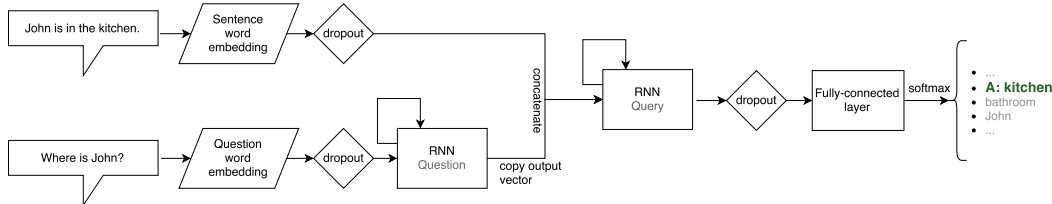

Figure 2: Model architecture for the bAbI dataset. The RNN is either LSTM or RotLSTM.

Table 1: Performance comparison on the bAbI dataset. Values are % average accuracy on test set ±standard deviation taken from training each model 10 times. For each of the trained models the test accuracy is taken for the epoch with the best validation accuracy (picked from epoch numbers 1, 11, 21, 31, 40 since we only evaluated on the test set for those).

| Task: | 1 | 2 | 3 | 4 | 5 | 6 | 7 | 8 | 9 | 10 |
|---|---|---|---|---|---|---|---|---|---|---|
| **LSTM** | 49.8 ±3.3 | 26.5 ±3.8 | 21.1 ±1.0 | 63.7 ±9.4 | 33.6 ±6.2 | 49.2 ±0.8 | 62.7 ±15.5 | **68.8 ±8.3** | **63.9 ±0.2** | 45.2 ±2.0 |
| **RotLSTM** | **52.3 ±1.1** | **27.1 ±1.3** | **22.4 ±1.3** | **65.8 ±3.6** | **55.7 ±5.5** | **50.1 ±1.9** | **76.7 ±3.0** | 66.1 ±6.2 | 61.5 ±1.9 | **48.1 ±1.1** |

| Task: | 11 | 12 | 13 | 14 | 15 | 16 | 17 | 18 | 19 | 20 |
|---|---|---|---|---|---|---|---|---|---|---|
| **LSTM** | 73.6 ±2.2 | 74.3 ±1.7 | 94.4 ±0.2 | **20.7 ±3.3** | 21.4 ±0.5 | **48.0 ±2.2** | 48.0 ±0.0 | 70.3 ±20.8 | 8.5 ±0.6 | 87.7 ±4.2 |
| **RotLSTM** | **73.9 ±1.2** | **76.5 ±1.1** | **94.4 ±0.0** | 19.9 ±2.0 | **28.8 ±8.8** | 46.7 ±2.0 | **54.2 ±3.1** | **90.5 ±0.9** | **8.9 ±1.1** | **89.9 ±2.4** |

The categorical cross-entropy loss function was used for training. All dropout layers are dropping 30% of the nodes. The train-validation dataset split used was 95%-5%. The optimizer used was Adam with learning rate 0.001, no decay, $\beta_1 = 0.9$, $\beta_2 = 0.999$, $\epsilon = 10^{-8}$. The training set was randomly shuffled before every epoch. All models were trained for 40 epochs. After every epoch the model performance was evaluated on the validation and training sets, and every 10 epochs on the test set. We set the random seeds to the same number for reproducibility and ran the experiments 10 times with 10 different random seeds. The source code is available at `https://goo.gl/Eopz2C`[2].

## 3.2 RESULTS

In this subsection we compare the the performance of models based on the LSTM and RotLSTM units on the bAbI dataset.

Applying rotations on the unit memory of the LSTM cell gives a slight improvement in performance overall, and significant improvements on specific tasks. Results are shown in Table 1. The most significant improvements are faster convergence, as shown in Figure 3, and requiring smaller state sizes, illustrated in Figure 4.

On tasks 1 (basic factoid), 11 (basic coreference), 12 (conjunction) and 13 (compound coreference) the RotLSTM model reaches top performance a couple of epochs before the LSTM model consistently. The RotLSTM model also needs a smaller cell state size, reaching top performance at state size 10 to 20 where the LSTM needs 20 to 30. The top performance is, however, similar for both models, with RotLSTM improving the accuracy with up to 2.5%.

The effect is observed on task 18 (reasoning about size) at a greater magnitude where the RotLSTM reaches top performance before epoch 20, after which it plateaus, while the LSTM model takes 40 epochs to fit the data. The training is more stable for RotLSTM and the final accuracy is improved by 20%. The RotLSTM reaches top performance using cell state 10 and the LSTM needs size 40. Similar performance increase for the RotLSTM (22.1%) is observed in task 5 (three argument relations), reaching top performance around epoch 25 and using a cell state of 50. Task 7 (counting) shows a similar behaviour with an accuracy increase of 14% for RotLSTM.

Tasks 4 (two argument relations) and 20 (agent motivation) show quicker learning (better performance in the early epochs) for the RotLSTM model but both models reach their top performance

---

[2]A GitHub repository will be made available after the double-blind review period ends.

after the same amount of traning. On task 20 the RotLSTM performance reaches top accuracy using state size 10 while the LSTM incremetally improves until using state size 40 to 50.

Signs of overfitting for the RotLSTM model can be observed more prominently than for the LSTM model on tasks 15 (basic deduction) and 17 (positional reasoning).

Our models, both LSTM and RotLSTM, perform poorly on tasks 2 and 3 (factoid questions with 2 and 3 supporting facts, respectively) and 14 (time manipulation). These problem classes are solved very well using models that look over the input data many times and use an attention mechanism that allows the model to focus on the relevant input sentences to answer a question (Sukhbaatar et al., 2015; Kumar et al., 2016). Our models only look at the input data once and we do not filter out irrelevant information.

## 4    DISCUSSION AND FUTURE WORK

A limitation of the models in our experiments is only applying pairwise 2D rotations. Representations of past input can be larger groups of the cell state vector, thus 2D rotations might not fully exploit the benefits of transformations. In the future we hope to explore rotating groups of elements and multi-dimensional rotations. Rotating groups of elements of the cell state could potentially also force the models to learn a more structured representation of the world, similar to how forcing a model to learn specific representations of scenes, as presented in Higgins et al. (2017), yields semantic representations of the scene.

Rotations also need not be fully flexible. Introducing hard constraints on the rotations and what groups of parameters can be rotated might lead the model to learn richer memory representations. Future work could explore how adding such constraints impacts learning times and final performance on different datasets, but also look at what constraints can qualitatively improve the representation of long-term dependencies.

In this work we presented preliminary tests for adding rotations to simple models but we only used a toy dataset. The bAbI dataset has certain advantages such as being small thus easy to train many models on a single machine, not having noise as it is generated from a simulation, and having a wide range of tasks of various difficulties. However it is a toy dataset that has a very limited vocabulary and lacks the complexity of real world datasets (noise, inconsistencies, larger vocabularies, more complex language constructs, and so on). Another limitation of our evaluation is only using text, specifically question answering. To fully evaluate the idea of adding rotations to memory cells, in the future, we aim to look into incorporating our rotations on different domains and tasks including speech to text, translation, language generation, stock prices, and other common problems using real world datasets.

Tuning the hyperparameters of the rotation models might give better insights and performance increases and is something we aim to incorporate in our training pipeline in the future.

A brief exploration of the angles produced by $\mathbf{u}$ and the weight matrix $\mathbf{W}_{\text{rot}}$ show that $\mathbf{u}$ does not saturate, thus rotations are in fact applied to our cell states and do not converge to 0 (or 360 degress). A more in-depth qualitative analysis of the rotation gate is planned for future work. Peeking into the activations of our rotation gates could help understand the behaviour of rotations and to what extent they help better represent long-term memory.

A very successful and popular mutation of the LSTM is the Gated Recurrent Unit (GRU) unit (Cho et al., 2014). The GRU only has an output as opposed to both a cell state and an output and uses fewer gates. In the future we hope to explore adding rotations to GRU units and whether we can obtain similar results.

## 5    CONCLUSION

We have introduced a novel gating mechanism for RNN units that enables applying a parametrised transformation matrix to the cell state. We picked pairwise 2D rotations as the transformation and shown how this can be added to the popular LSTM units to create what we call RotLSTM.

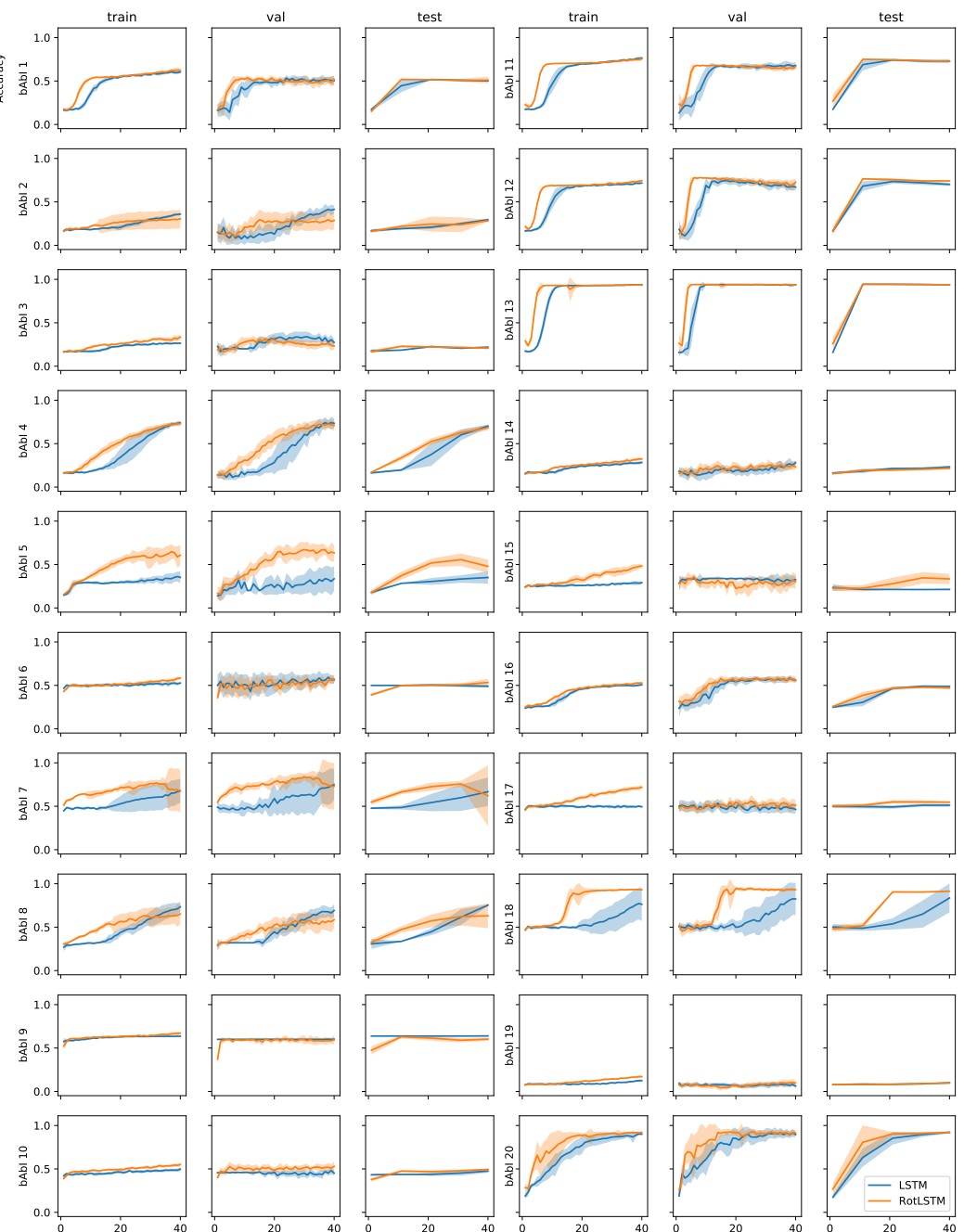

Figure 3: Accuracy comparison on training, validation (val) and test sets over 40 epochs for LSTM and RotLSTM models. The models were trained 10 times and shown is the average accuracy and in faded colour is the standard deviation. Test set accuracy was computed every 10 epochs.

We trained a simple model using RotLSTM units and compared them with the same model based on LSTM units. We show that for the LSTM-based architectures adding rotations has a positive impact on most bAbI tasks, making the training require fewer epochs to achieve similar or higher accuracy. On some tasks the RotLSTM model can use a lower dimensional cell state vector and maintain its performance.

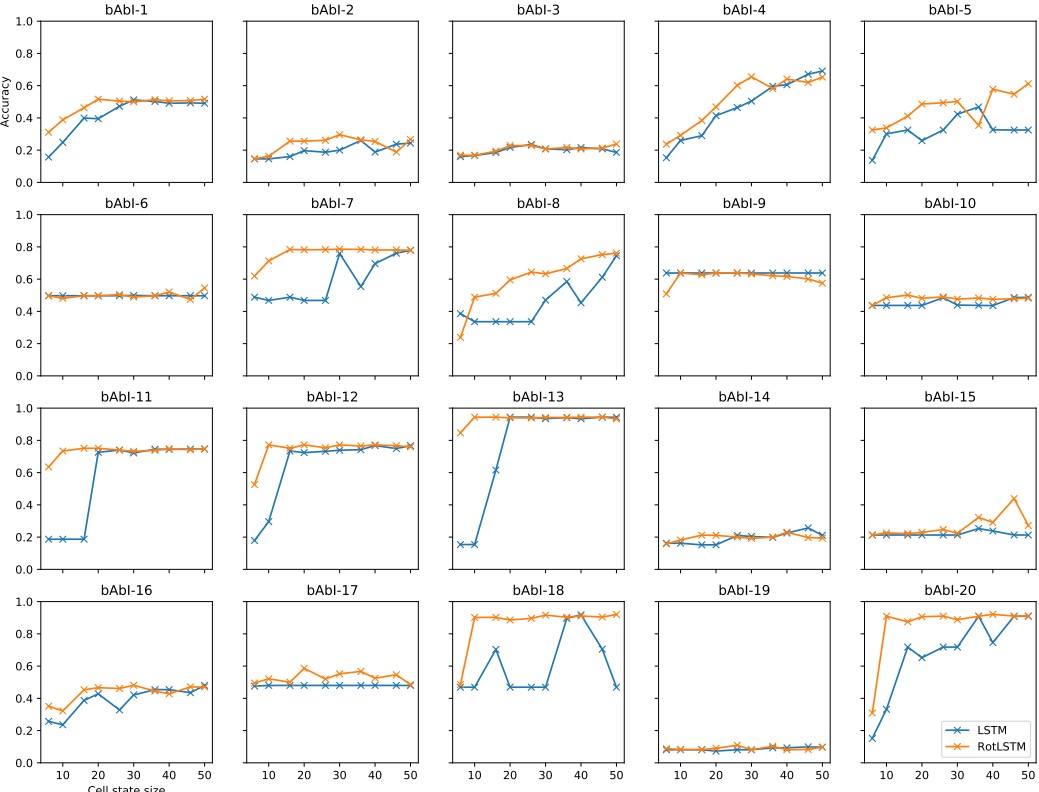

Figure 4: Accuracy on the test set for the LSTM and RotLSTM while varying the cell state size from 6 to 50. The shown numbers are for the epochs with best validation set accuracy.

Significant accracy improvements of approximatively 20% for the RotLSTM model over the LSTM model are visible on bAbI tasks 5 (three argument relations) and 18 (reasoning about size).

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

## A   BABI TASKS

A summary of all bAbI tasks is shown in Table 2 for reference.

Table 2: Descriptions of bAbI dataset tasks.

| # | Description | # | Description |
|---|---|---|---|
| 1 | Factoid QA with one supporting fact | 11 | Basic coreference |
| 2 | Factoid QA with two supporting facts | 12 | Conjunction |
| 3 | Factoid QA with three supporting facts | 13 | Compound coreference |
| 4 | Two argument relations: subject vs. object | 14 | Time manipulation |
| 5 | Three argument relations | 15 | Basic deduction |
| 6 | Yes/No questions | 16 | Basic induction |
| 7 | Counting | 17 | Positional reasoning |
| 8 | Lists/Sets | 18 | Reasoning about size |
| 9 | Simple Negation | 19 | Path finding |
| 10 | Indefinite Knowledge | 20 | Reasoning about agent's motivation |

