# OpenReview forum: "Modifying memories in a Recurrent Neural Network Unit"
_ICLR.cc/2018/Conference — Reject_

### Official Review · AnonReviewer3 · 2017-11-27
**Insufficient Justification and Comparison**

**Rating:** 4
**Confidence:** 3

**Review:**

The paper proposes to add a rotation operation in long short-term memory (LSTM) cells. It performs experiments on bAbI tasks and showed that the results are better than the simple baselines with original LSTM cells. There are a few problems with the paper.

Firstly, the title and abstract discuss "modifying memories", but the content is only about a rotation operation. Perhaps the title should be "Rotation Operation in Long Short-Term Memory"?

Secondly, the motivation of adding the rotation operation is not properly justified. What does it do that a usual LSTM cell could not learn? Does it reduce the excess representational power compared to the LSTM cell that could result in better models? Or does it increase its representational capacity so that some pattern is modeled in the new cell structure that was not possible before? This is not clear at all after reading the paper. Besides, the idea of using a rotation operation in recurrent networks has been explored before [3].

Finally, the task (bAbI) and baseline models (LSTM from a Keras tutorial) are too weak. There have been recent works that nearly solved the bAbI tasks to perfection (e.g., [1][2][4][5], and many others). The paper presented a solution that is weak compared to these recent results.

In a summary, the main idea of adding rotation to LSTM cells is not properly justified in the paper, and the results presented are quite weak for publication in ICLR 2018.

[1] Sainbayar Sukhbaatar, Jason Weston, Rob Fergus. End-to-end memory networks, NIPS 2015
[2] Caiming Xiong, Stephen Merity, Richard Socher. Dynamic Memory Networks for Visual and Textual Question Answering, ICML 2016
[3] Mikael Henaff, Arthur Szlam, Yann LeCun, Recurrent Orthogonal Networks and Long-Memory Tasks, ICML 2016
[4] Caglar Gulcehre, Sarath Chandar, Kyunghyun Cho, Yoshua Bengio, Dynamic Neural Turing Machine with Soft and Hard Addressing Schemes, ICLR 2017
[5] Mikael Henaff, Jason Weston, Arthur Szlam, Antoine Bordes, Yann LeCun, Tracking the World State with Recurrent Entity Networks, ICLR 2017

---

### Official Review · AnonReviewer1 · 2017-11-27
**Insufficient motivation and experimental analysis**

**Rating:** 4
**Confidence:** 3

**Review:**

The paper proposes an additional transform in the recurrent neural network units. The transform allows for explicit rotations and swaps of the hidden cell dimensions. The idea is illustrated for LSTM units, where the transform is applied after the cell values are computed via the typical LSTM updates.

My first concern is the motivation. I think the paper needs a more compelling example where swaps and rotations are needed and cannot otherwise be handled via gates. In the proposed example, it's not clear to me why the gate is expected to be saturated at every time step such that it would require the memory swaps. Alternatively, experimentally showing that the network makes use of swaps in an interpretable way (e.g. at certain sentence positions) could strengthen the motivation.

Secondly, the experimental analysis is not very extensive. The method is only evaluated on the bAbI QA dataset, which is a synthetic dataset. I think a language modeling benchmark and/or a larger scale question answering dataset should be considered.

Regarding the experimental setup, how are the hyper-parameters for the baseline tuned? Have you considered training jointly (across the tasks) as well?

Also, is the setting the same as in Weston et al (2015)? While for many tasks the numbers reported by Weston et al (2015) and the ones reported here for the LSTM baseline are aligned in the order of magnitude, suggesting that some tasks are easier or more difficult for LSTMs, there are large differences in other cases, for task #5 (here 33.6, Weston 70), for task #16 (here 48, Weston 23), and so on.

Finally, do you have an intuition (w.r.t. to swaps and rotations) regarding the accuracy improvements on tasks #5 and #18?

Some minor issues:
- The references are somewhat inconsistent in style: some have urls, others do not; some have missing authors, ending with "et al".
- Section 1, second paragraph: senstence
- Section 3.1, first paragraph: thorugh
- Section 5: architetures

---

### Official Review · AnonReviewer2 · 2017-11-28
**This paper is not ready for the publication**

**Rating:** 3
**Confidence:** 4

**Review:**

Summary: This paper introduces a model that combines the rotation matrices with the LSTMs. They apply the rotations before the final tanh activation of the LSTM and before applying the output gate. The rotation matrix is a block-diagonal one where each block is a 2x2 rotations and those rotations are parametrized by another neural network that predicts the angle of the rotations. The paper only provides results on the bAbI task.

Questions:
Have you compared against to the other parametrizations of the LSTMs and rotation matrices? (ablation study)
Have you tried on other tasks?
Why did you just apply the rotations only on d_{t}.

Pros:
Uses a simple parametrization of the rotation matrices.

Cons:
Not clear justification and motivations
The experiments are really lacking:
No ablation study
The results are only limited to single toy task.


General Comments:

This paper proposes to use the rotation matrices with LSTMs. However there is no clear justification why is this particular parametrization of rotation matrix is being used over others and why is it only applied before the output gate. The experiments are seriously lacking, an ablation study should have been made and the results are not good enough. The experiments are only limited to bAbI task which doesn’t tell you much. This paper is not ready for publication, and really feels like it is rushed.

Minor Comment:
This paper needs more proper proof-reading. There are some typos in it, e.g.:
1st page, senstence --> sentence
4th page, the the ... --> the

---

### Public Comment · ~Jack_William_Rae1 · 2017-11-03
**LSTM baseline**

It is worth noting there is a more recent bAbI LSTM baseline that outperforms the proposed model for almost all of the tasks:

https://arxiv.org/abs/1610.09027

Table 2 (in the suppl.). It may be worth comparing to these numbers. In that case the LSTM is jointly trained over all tasks, and is not tuned on a per-task basis. That was with a cell size of 100, trained with RMSProp with a learning rate of 1e-5.

---

### Public Comment · ~Andree_Kaba1 · 2017-12-16
**Reproducibility attempt**

The paper proposes the usage of a 2-D rotation based gating mechanism in LSTM (RotLSTM) to increase accuracy and speed up the convergence. The validation that was presented in the paper shows that the RotLSTM outperforms a baseline LSTM in most of the bAbI tasks and in some cases, it requires a smaller cell size to achieve the same accuracy and it converges earlier. As part of the reproducibility challenge, we downloaded the code that was provided with the paper and tried to reproduce the results.  We used that code to see if the numbers that were published matched the paper results. We also tried to see if the conclusion held after performing model selection and picking better models.

Pros:
The code was helpful, implementing exactly what was presented in the paper. It did not take us more than a couple of hours to reuse it for other purposes (model selection or different architectures and different datasets).
The ideas in the paper were clearly explained and the paper was easy to understand.

Cons:
When we rerun the published code on the same tasks with the same hyper-parameters we had a different outcome. In our setup RotLSTM did not converge faster than the baseline LSTM except on task 18, and accuracy wise it performed at most as well as the baseline LSTM.

We performed model selection to choose a good LSTM to compare with. We limited the tasks to task 5, task 7, and task 18 as they were the best-performing tasks from the published results. In 2 out of these three tasks (5 and 7), training the selected LSTM and a RotLSTM that uses the same hyperparameters shows that the RotLSTM performed better than the LSTM. In the third task (task 18) it was the opposite. The hyper-parameters that were tuned during this task are the epochs, the batch size, the embedded hidden size, the query hidden size and the sentence hidden size. We performed Bayesian optimization to select the model, each 5-uplet of values is used to train 5 different models and the output of the objective function is the average of accuracies of these 5 models. The regions of the search are:
Epochs [5:5:200] ([start:step:end])
Batch size [16:16:256]
Embedded hidden size, query hidden size, and sentence hidden size [1:1:100] each

Other comments:
The experiments took significant time to be performed. Model selection for each task takes between 3h30 and 4h00 (on 4 core blade using Keras and GPyOpt’s multicore options). The reproduction of the results regarding the impact of the cell size on the test accuracy took 24 hours of experimentation (4 cores and Tesla K80 GPU).
We tried to experiment with a time series dataset but we found no difference in accuracy between LSTM and RotLSTM. RotLSTM also required significantly more time to train.

Reproducibility report: https://www.dropbox.com/s/ok4z66ccg5m6bff/comp-551-final.pdf?dl=0

---

### Decision · Program_Chairs · 2018-01-29
**ICLR 2018 Conference Acceptance Decision**

**Decision:**

Reject

**Comment:**

the idea is interesting, but as pointed out the reviewers (and also agreed by the authors), the current manuscript lacks clear motivations, reasons underlying specific design choices and convincing empirical evaluation.